# Latrine utilization and associated factors among districts implementing and not-implementing community-led total sanitation and hygiene in East Wollega, Western Ethiopia: A comparative cross-sectional study

**Adisu Tafari Shama**[1]*, **Dufera Rikitu Terefa**[1], **Edosa Tesfaye Geta**[1], **Melese Chego Cheme**[1], **Bayise Biru**[1,2], **Jira Wakoya Feyisa**[1], **Matiyos Lema**[1], **Adisu Ewunetu Desisa**[1], **Bikila Regassa Feyisa**[1,3], **Dejene Seyoum Gebre**[1]

**1** Department of Public Health, Institute of Health Sciences, Wollega University, Nekemt, Ethiopia,
**2** Department of Human Nutrition and Dietetics, Faculty of Public Health, Jimma University, Jimma, Ethiopia,
**3** Department of Epidemiology, Faculty of Public Health, Jimma University, Jimma, Ethiopia

* adisuteferi1906@gmail.com

## Abstract

### Introduction

Discharge of excreta to the environment lead to surface and groundwater contamination and human exposure to disease-causing micro-organisms. There is limitation of evidences regarding the latrine utilization among community-led total sanitation and hygiene implemented and non-implemented districts of the East Wollega Zone. Hence, this study aimed to determine the magnitude and associated factors of latrine utilization among households in community-led total sanitation and hygiene implemented and non-implemented Districts in East Wollega Zone, Western Ethiopia.

### Methods

A cross-sectional study was conducted. A Multi-stage sampling technique was applied to select the 461 households. Data were collected using interviews and observations guided by a pre-structured questionnaire. Data were entered using Epi Data and exported to SPSS software version 25 for data recording, cleaning, and statistical analysis. Bivariable logistic regression was run to identify the candidate variables at p-value <0.25. Variables that had associations with latrine utilization in the bi-variable analysis were entered together into multivariable logistic regression. An Adjusted odds ratio with a 95% confidence interval was calculated and P-value< 0.05 was used to declare a statistically significant association.

### Result

The overall prevalence of latrine utilization was found to be 52.7% (95%CI:48%, 57.3%). Religion (AOR = 0.149;95%CI:0.044,0.506), education (AOR = 3.861;95%CI:1.642,9.077), occupation, absence of children <5 years (AOR = 4.724;95%CI:2.313,9.648), toilet cleaning

**Data Availability Statement:** All relevant data are within the paper and its Supporting Information files.

**Funding:** The authors received no specific funding for this work.

**Competing interests:** The authors have declared that no competing interests exist.

**Abbreviations:** CLTS, Community-led total sanitation; CLTSH, Community-led total sanitation and hygiene; HI, Health institution, NGOs: non-governmental organization; OD, open defecation; ODF, open defecation Free; SDG, Sustainable Development Goal; WASH, Water, Sanitation and Hygiene; VIPL, Ventilated improved pit latrine.

(AOR = 10.662;95%CI:5.571,20.403), frequency of latrine construction (AOR = 6.441;95% CI:2.203,18.826), maintenance need (AOR = 6.446; 95%CI:3.023,13.744), distance from health institution (AOR = 0.987; 95%CI:0.978, 0.996), distance from kebele office (AOR = 6.478; 95%CI:2.137,19.635), and latrine distance from dwelling (AOR = 11.656; 95% CI:2.108, 64.44) were the factors associated with latrine use.

## Conclusion

The latrine utilization in this study is low as compared to other studies. Religion, education, occupation, absence of children <5 years, toilet cleaning, frequency of latrine construction, maintenance need of the toilet, distance from health institution, distance from kebele, and latrine distance from dwelling are the associated factors of latrine utilization. Both households and health workers have to work together to improve latrine utilization and the safe disposal of children's feces.

## Introduction

Unsafe management and discharge of excreta to the environment expose human to disease causing-micro-organisms by causing surface and groundwater contamination [1, 2]. In other ways, a latrine is a facility that is used for the safe disposal of human faeces and urine [3]. It is the lowest cost option that ensures a clean and healthful living environment both at home and in the neighborhood of users [4]. The use of accessible improved latrines in households and other settings is the core strategy in water, sanitation, and hygiene (WASH) interventions and also to fight neglected tropical diseases which are affecting one billion people globally [5]. Further, effective utilization of latrine can prevent diarrhea and malnutrition [6].

There is confusion around the issue of how the accessibility of improved latrines is determined. Scholars argue that estimation of access to improved latrine needs to consider the chain of the faecal sludge management system from containment to adequate treatment as well as proper utilization and user behavior not merely the type of latrine technology. Hence, there is a need to consider adequate treatment and valorization of fecal wastes as a criterion for using improved latrines in the Sustainable development goal (SDG) era [7].

To improve latrine ownership and utilization, various approaches have been implemented by government and non-governmental organizations. For instance, community-led total sanitation (CLTS), pioneered by Dr. Kamal Kar, is one of the new approaches implemented to reduce open defecation (OD) and improve hygiene and sanitation practices in different parts of developing countries. Study findings showed that CLTS was an important approach for increasing latrine ownership and utilization rates [8–12]. The Ethiopian Government added the hygiene component and adapted the approach by renaming it as community-led total sanitation and hygiene (CLTSH) in 2006 to improve the sanitation and hygiene status of the community [13]. The focus of CLTSH is the eradication of open defecation (OD) at a community level by generating sustained change in the collective behavior of OD practices and initiating latrine demand without outer support [14–16]. The communities are enabled to observe, appraise, and analyze their OD practice and its effects. The absence of the provision of sanitation facilities is the characteristic of this CLTS [14, 17].

In the SDG, United Nations has planned to end open defecation by paying special attention to those in need [18]. Globally, 2.4 billion population lack sanitary facilities, and about one

billion practice open defecation [19]. About 28.5% of the Ethiopian population is still practicing open defecation with the highest proportion in rural than urban areas (35.4% versus 10.1%) [20] and one out of six Ethiopian households is slipping into open defecation practices after certification for ODF [21]. The studies conducted in various areas of Ethiopia revealed a 31.08%-90% prevalence of latrine utilization [22–26]. This level of latrine utilization in Ethiopia is poor compared to the sustainable development goal target of ensuring 100% latrine ownership and utilization [18]. Moreover, a significant proportion (56%) of rural populations is utilizing unimproved latrines which do not guarantee breaking the chain of contact with human feces and urine- the purpose of sanitation [20].

Poor latrine utilization conditions are among the major causes of public health problems in Ethiopia where children are the most vulnerable [27]. Communicable diseases account for about 60% to 80% of health problems that are preventable and considerable proportions of these diseases are directly related to poor latrine utilization [28, 29]. Unsanitary disposal of human excreta, together with unsafe drinking water and poor hygiene conditions contribute to 88% of diarrheal diseases. In addition, inadequate sanitation is implicated in helminthic infections, enteric fevers, and trachoma. The burden of this disease is a leading cause of morbidity and mortality, particularly in young children. Poor latrine utilization results in 5 to 12 episodes of diarrhea and between 50, 000 to 112,000 under-five children die annually in Ethiopia. Lack of access to sanitation has significant non-health consequences, especially for women and girls, including lack of security and privacy, decreased school attendance, and basic human dignity [28, 30–34]. Economically, poor sanitation costs 13.5 billion Ethiopian Birr each year (Water safety plan. Desk review on Economics of Sanitation (ESI) for Ethiopia. 2015; unpublished).

So far conducted studies identified that latrine utilization is affected by socio-demographic and economic, behavioral, and environmental factors [22–26].

To the authors' knowledge, even though there are studies conducted on latrine utilization in different regions of Ethiopia, there is a scarcity of the study to compare the prevalence of latrine utilization among villages in CLTSH-implemented and non-implemented districts particularly in the study area. Besides, the status of CLTSH implementation and challenges are not well addressed in many of the previous studies. Therefore, this study was intended to compare the prevalence of latrine utilization and associated factors among the CLTSH-implemented and non-implemented districts of East Wollega, Western Ethiopia.

## Methods and materials

### Study area and period

The study was conducted in the East Wollega Zone, Oromia Regional State of Ethiopia from July 1 to 30, 2022. East Wollega is one of the 21 Zones found in the Oromia Regional State. It is located 331 km far from Addis Ababa, the capital city of Ethiopia. East Wollega zone has 17 Districts, 340 kebeles (the smallest administrative unit in the Ethiopian context), 338,022 households, and a 1,622,507 total population. Although CLTSH was implemented in all districts, it didn't reach some villages with variation from district to district. Accordingly, only one District (Gobu Sayo) has implemented the CLTSH with better reach for many of the villages while the other districts didn't reach many of their villages (East Wollega Zonal health department Annual performance Review. August 2021, Farmland Hotel, Nekemte, Oromia, Ethiopia. Unpublished).

### Study design

A comparative community-based cross-sectional study design was employed.

## Source population

All households in the East Wollega zone were the source population for the study.

**Study population.** All the households that have a private latrine and were found in the selected districts of East Wollega were the study population.

## Inclusion and exclusion criteria

**Inclusion criteria.** The study inclusion criteria were all households who resided in the area for at least six months.

**Exclusion criteria.** All households who satisfied the inclusion criteria but who were away from home and/or houses that were closed or could not be accessed at the time of the survey for two visits were excluded from the study.

## Sample size determination

The study sample size was calculated by using a double population proportion formula. The proportions of latrine utilization in CLTS implemented districts (P1 = 54.9%) and proportion in non-CLTS districts (P2 = 38.7%) were considered from the study done in the Laelai Mai-chew district, Tigray, North Ethiopia [35]. Assuming the 95% level of confidence, 80% power, 1 to 1 ratio, 10% non-response rate, and design effect = 1.5, the required sample size n was: -

$$n = \frac{2*[(Zcv\sqrt{2p(1-p)}+Zpower\sqrt{P1(1-P1)+P2(1-P2)})^2]}{(P1-P2)^2} + 10\% \text{ non-response rate,}$$

Whereas, n = the sample size

Zcv = Z critical value for alpha which is 1.96 for this case, Zpower = Z value for 1-beta which is 0.842 in this study

P1 = expected proportion of latrine utilization in sample one, P2 = expected proportion of latrine utilization for sample two, $p = \frac{P1+P2}{2}$

This yielded a total of 325 households. Then after multiplying by 1.5 for design effect, 488 households (244 households at each site) were considered to be included in the study.

## Sampling procedure/technique

The multi-stage sampling technique was applied to select the households for the study on adjusting for design effect. First, one district that had better the implementation of CLTSH and another district that did not reach many of its villages and implemented CLTSH were randomly selected by lottery method from the Zone. Then six Kebeles three from each CLTSH implemented and non-implemented districts were selected randomly by lottery method. From each kebeles, 3 villages were selected. The calculated sample size was proportionally allocated to the selected villages. To draw a sampling frame, the total number of households in the villages was obtained from the respective village leader (*abbaa or haadha garee*). Finally, using a systematic random sampling technique (every k$^{th}$ number of households), the final sample size included in the study was selected.

## Data collection methods

The data were collected through face-to-face interviews and observations using a pre-structured standardized questionnaire adapted from the world health organization and the united nations children's fund [36]. The questionnaire comprised four sections: socio-demographic and economic characteristics, behavioral factors, environmental factors, and latrine utilization-related questions that were supported by the observation. Nine first-degree holder health

professionals and three Master's degree holder health workers who can speak the Afaan Oromo language were recruited as data collectors and supervisors, respectively.

**Study variables.**   *Dependent variable*. Latrine utilization.

*Independent variables*. **Socio-demography variables**: Age, marital status, family size, educational status, presence of educated children in the household, occupation, family income, and presence of under-five children.

*Behavioral variables***:** frequency of cleaning, hygienic condition of latrine, knowledge about the importance of latrine use, and reasons for constructing latrine.

**Environmental Variables:** latrine service year, presence and type of slab, presence of the door, frequency of latrine construction, the need of maintenance, number of households using well-constructed slab and superstructure latrine, implementation of CLTSH, latrine distance from the dwelling, duration of latrine owned, frequency of visit by health workers, hand washing facility and distance from health institution (HI)/kebele offices.

## Operational definitions

**Latrine Utilization** was determined by assessing the presence of functional and improved latrines, safe disposal of children's feces, absence of observable feces in the compound, and at least one observable sign of use (the footpath to the latrine is not covered by grass, latrine has an odor, lack of spider web in squatting hole, presence of anal cleansing material, fresh faeces in the squatting hole, or a wet slab) [37, 38].

**Functional latrines:** Latrines that were found providing services (unblocked holes/not full/unbroken slabs, and accessible to the families) during the data collection period [39].

**Improved latrines:** Latrines such as flush/pour flush latrine connected to either sewer system, septic tanks, or pit latrine, ventilated improved pit latrine (VIPL), pit latrine with slab, and composting latrine were considered as improved latrines [40].

**Safe disposal of children's faeces:** This variable was assessed for children less than five years and it was considered as 'safely disposed of' when either the child used the toilet or the family disposed of the child's faeces in a toilet or buried it [41].

**Clean latrine**: This is the absence of fecal matter around the pit latrine and the squat hole, and properly swept latrine. It was measured based on the report of the household regarding the toilet cleaning plus observation of the feces around the squat hole and toilet.

## Data quality control

After preparing the English version of the questionnaire, it was translated into the Afaan Oromo language and then back-translated into English by another person to ensure that the originality and meaning were retained. The pre-test of the questionnaire was done on 5% of the sample in the non-selected district to identify any ambiguity, consistency, and acceptability of the questionnaire. Then necessary corrections were made before the actual data collection to make it ready for final data collection.

Two days of training were given to the data collectors and supervisors on the topic, how to collect the data, when and how to make an interview, and ethical issues emphasizing the importance of the safety of participants and data quality. In addition, the quality of data was monitored frequently both in the field and during data entry. This was done in the field through close supervision of interviewers. Data quality tables were utilized. All completed questionnaires were examined for completeness and consistency during the interview. An incomplete and unclearly filled questionnaire was given back to the interviewer, and got complete questionnaires each day. Double data entry using a programmed computer software package was done by two individuals.

## Data processing and analysis

The data were entered into Epi Data and exported to SPSS software version 25 for data recording, cleaning, and statistical analysis. Descriptive statics using frequencies, percentages, tables, and figures were used to describe the magnitude of latrine utilization among households. Bivariable logistic regression analysis was done to identify variables that were candidates for multivariable analysis. All variables that had an association on bivariable analysis at p-value <0.25 were considered for inclusion in the multivariate analysis. Then multivariable analysis was done to control the confounding effect of other variables and to identify independent factors of latrine utilization. A backward selection method was used to select the variables. Multi-collinearity was also checked using the correlation matrix with the result of no collinearity problem detected. Model fitness was checked using Hosmer and Lemeshow's Test and the result showed a p-value of 0.083 which indicated that the model fitted the data. The magnitude and direction of the relationship between the variables were expressed as odds ratios (OR) and p-value < 0.05 was used to declare the statistically significant associations. Missing data were excluded from the analysis.

## Ethical considerations

This research was done according to the Helsinki declaration regarding the research ethical issues. Ethical clearance was obtained from the Wollega University, research ethics review committee before data collection with the reference number IHSRPTTAD90/2014. Institute of Health Sciences wrote the official letter to the East Wollega health department. Also, letter of cooperation was taken from the zonal health department and oral permission was obtained from the district Health offices before the start of data collection. All necessary measures were made to guard against any form of harm and discomfort to the study subjects. The study purpose, risks, and benefits were explained in the local language for the participants. Their informed, voluntary, written and signed consent, in the end, was sought and participants thumb printed consent to participate in the study. Confidentiality was also guarded by making sure that the study participants will not be represented by their names. In addition, password protection of soft data and the use of a key and lock for hard copy data was employed to guarantee confidentiality.

## Result

Of the 488, 461 households (229 from the villages in CLTSH-implemented versus 232 from the CLTSH non-implemented districts) participated in this study with a response rate of 94.5%. The mean age with standard deviation (SD) of the respondent was 44.53 ±11.36 years. The average family size of the household was 5.34± 1.94 SD persons. The average walking time from home to health institutions was 73.64 ±42.704 SD minutes (Table 1).

About half of the households-49.3% in CLTSH implemented villages and 50.4%% in CLTSH non-implemented villages clean their toilet. The majority of the households 149 (65.1%) in CLTSH implemented villages and 143 (61.6%) in CLTSH non-implemented villages were motivated to construct the toilet by primary health workers. Feces were observed in about one-third-34.9% of compounds of the study households while majority-427 (92.6%) of the households know the disease preventability of latrine utilization (Table 2).

More than three quarters-365 (79.2%) of the households have hand washing facilities near their toilet. About 379 (82.2%) of the households in the study reported that they wash their hands after the toilet visits. The average water consumption for hand washing in a day was 6.67 ±11.35 SD liters among the households (Fig 1).

**Table 1. Socio-demographic and economic characteristics of the households that involved in the study of latrine utilization, 2022 (461).**

| Variable category | | CLTSH ever implemented | |
|---|---|---|---|
| | | Yes (n = 229); frequency (%) | No (n = 232); frequency (%) |
| Age in years | 18–29 | 10 (4.4) | 16 (6.9) |
| | 30–39 | 66 (28.8) | 65(28.0) |
| | 40–49 | 79 (34.5) | 73 (31.5) |
| | 50–59 | 43 (18.8) | 42 (18.1) |
| | > = 60 | 31 (13.5) | 36 (15.5) |
| Gender | Female | 35 (15.3) | 25 (10.8) |
| | Male | 194 (84.7) | 207 (89.2) |
| Religion | Protestant | 75 (32.8) | 76 (32.8) |
| | Orthodox | 133 (58.1) | 141 (60.8) |
| | Catholic | 9 (3.9) | 10 (4.3) |
| | Muslim | 10 (4.4) | 5 (2.2) |
| | Other | 2 (0.9) | 0 (0) |
| Marital status | Single | 12 (5.2) | 11 (4.7) |
| | Widowed | 19 (8.3) | 11 (4.7) |
| | Divorced/Separated | 19 (8.3) | 19 (8.2) |
| | Married | 179 (78.2) | 191 (82.3) |
| Educational status of the respondent | Cannot read and write | 72 (31.4) | 68 (29.3) |
| | can read and write | 52 (22.7) | 69 (29.7) |
| | Primary school (1–8) | 84 (36.7) | 78 (33.6) |
| | High school (9–12) | 14 (6.1%) | 9 (3.9) |
| | college/university | 7 (3.1%) | 8 (3.4) |
| Educational status of the mother | Cannot read and write | 90 (39.3) | 96 (41.4) |
| | can read and write | 59 (25.8) | 78 (33.6) |
| | Primary school (1–8) | 66 (28.8) | 50 (21.6) |
| | High school (9–12) | 9 (3.9) | 5 (2.2) |
| | college/university | 5 (2.2) | 3 (1.3) |
| Have school children/educated children | No | 65 (28.4) | 60 (25.9) |
| | Yes | 164 (71.6) | 172 (74.1) |
| Occupation of household head | Farmer | 202 (88.2) | 207 (89.2) |
| | daily laborer | 12 (5.2) | 11 (4.7%) |
| | Merchant | 7 (3.1) | 5 (2.2%) |
| | gov employee | 5 (2.2) | 6 (2.6) |
| | NGO employee | 3 (1.3) | 3 (1.3) |
| Occupation of the mother | Farmer | 119 (52.0) | 120 (51.7) |
| | Housewife | 93 (40.6) | 93 (40.1) |
| | daily laborer | 7 (3.1) | 8 (3.4) |
| | Merchant | 2 (0.9) | 0 (0) |
| | gov employee | 5 (2.2) | 8 (3.4) |
| | NGO employee | 2 (0.9) | 2 (0.9) |
| | Others | 1 (0.4) | 1 (0.4) |
| Income in Ethiopian Birr | < = 1000 | 95 (41.5) | 83 (35.8) |
| | 1001–2000 | 50 (21.8) | 46 (19.8) |
| | 2001–3000 | 36 (15.7) | 44 (19) |
| | 3001–4000 | 31 (13.5) | 32 (13.8) |
| | 4001–5000 | 13 (5.7) | 19 (8.2) |
| | >5000 | 4 (1.7) | 8 (3.4) |

(*Continued*)

**Table 1.** (Continued)

| Variable category | | CLTSH ever implemented | |
|---|---|---|---|
| | | Yes (n = 229); frequency (%) | No (n = 232); frequency (%) |
| Have children <5 years | No | 111 (48.5) | 106 (45.7) |
| | Yes | 118 (51.5) | 126 (54.3) |
| Family size | ≤5 | 131 (57.2) | 139 (59.9) |
| | >5 | 98 (42.8) | 93 (40.1) |

Of the 365 households that have hand washing facilities, water is observed in the hand washing facilities of 343 (93.97%) households. Similarly, detergents are observed for 207 (56.7%) households having hand-washing facilities (Fig 2).

More than half-54.1% of toilets in CLTSH-implemented villages and about two-third-66.8% of toilets in CLTSH no-implemented villages have no door. Majority of the toilets; 82.5% in CLTSH implemented villages and 83.2% in CLTSH non-implemented villages have slabs made of soil and/or wood. About 134 (58.5%) households in CLTSH-implemented villages and 134 (57.8%) in CLTSH non-implemented villages constructed toilets for the second time. Regarding the maintenance need of a toilet, about two-thirds of the toilets;147 (64.2%) in CLTSH implemented villages, 160 (69.0%) in CLTSH non-implemented villages and 307 (66.6%) of the total sample households reported that their toilets need maintenance. Nearly two-thirds-302 (65.5%) of the households reported that they are located at a medium distance from the kebele office (Table 3).

**Table 2. Behavioral factors of latrine utilization in East Wollega, Western Ethiopia, 2022 (n = 461).**

| Variable category | | CLTSH ever implemented | | Overall Freq. (%) |
|---|---|---|---|---|
| | | Yes (n = 229); frequency (%) | No (n = 232); frequency (%) | |
| Clean toilet | Not clean | 116 (50.7) | 115 (49.6) | 231 (50.1) |
| | Clean | 113 (49.3) | 117 (50.4) | 230 (49.9) |
| Frequency of cleaning | Daily | 11 (4.8) | 13 (5.6) | 24 (5.2) |
| | Sometimes | 100 (43.7) | 117 (50.4) | 217 (47.1) |
| | Rarely | 71 (31.0) | 57 (24.6) | 128 (27.8) |
| | Do not clean at all | 47 (20.5) | 45 (19.4) | 92 (20.0) |
| Motivation/reason to construct latrine | Self-initiation | 28 (12.2) | 30 (12.9) | 58 (12.6) |
| | Kebele leaders | 50 (21.8) | 59 (25.4) | 109 (23.6) |
| | Primary health workers | 149 (65.1) | 143 (61.6) | 292 (63.3) |
| | Peers | 2 (0.9) | 0 (0) | 2 (0.4) |
| Observable feces in the compound | No | 152 (66.4) | 148 (63.8) | 300 (65.1) |
| | Yes | 77 (33.6) | 84 (36.2) | 161 (34.9) |
| Knowledge about disease preventability of latrine usage | No | 20 (8.7) | 14 (6.0) | 34 (7.4) |
| | Yes | 209 (91.3) | 218 (94.0) | 427 (92.6) |
| Know preventability by avoiding the probability of feces contact with fingers | | 115 (55) | 133 (61) | 248 (58.0) |
| Know preventability by avoiding contact of feces with fluids | | 108 (51.6) | 109 (50) | 217 (50.8) |
| Know preventability by avoiding contact of feces with fomites/materials | | 43 (205) | 29 (13.3) | 72 (16.9) |
| Know preventability by avoiding contact of feces with flies | | 36 (17.22) | 27 (12.3) | 63 (14.7) |
| Know preventability by avoiding contact of feces with food | | 79 (37.79) | 87 (39.9) | 166 (38.9) |

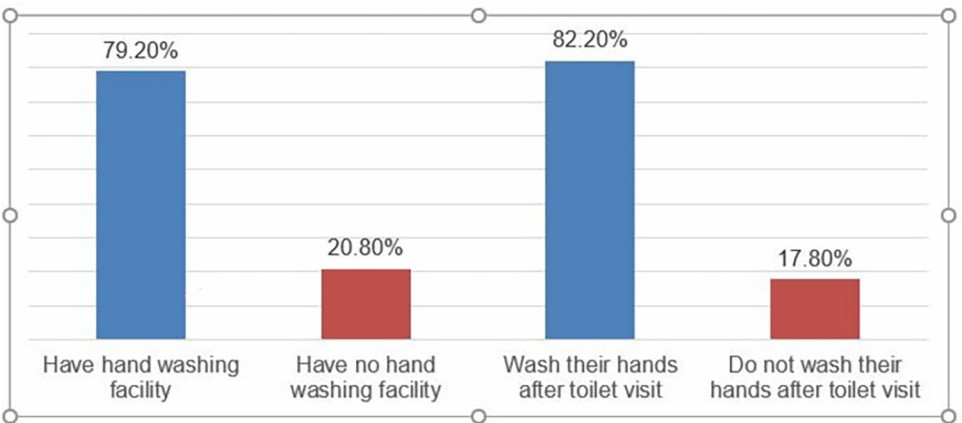

**Fig 1. Hand washing facility and hand washing practice of the study households, 2022 (n = 461).**

The majority 402 (87.2%) of households in this study has improved latrines (VIPL-156 (33.7%), pit latrine with slab-246 (53.7%)) whereas the rest 59 (12.8%) have unimproved latrines. The prevalence of latrine utilization in CLTSH implemented district was 123 (53.7% [95%CI:47%, 60.3%]) while it was 120 (51.7% [95%CI:45.1%, 58.3%]) in CLTSH non-implemented district (Fig 3).

Overall, more than half-243 (52.7% [(95%CI:48%, 57.3%]) of the households utilized their latrine (Fig 4).

In bivariable logistic regression, 12 variables showed a significant association with latrine utilization in CLTSH non-implemented districts. These variables are age 36–40 as compared to 18–25 years, family size, income, having of children <5 years, occupation of household head, cleaning toilet, motivation/reason to construct latrine, frequency of latrine construction, maintenance need, distance from health institution in minutes, distance from kebele office and hand washing facility.

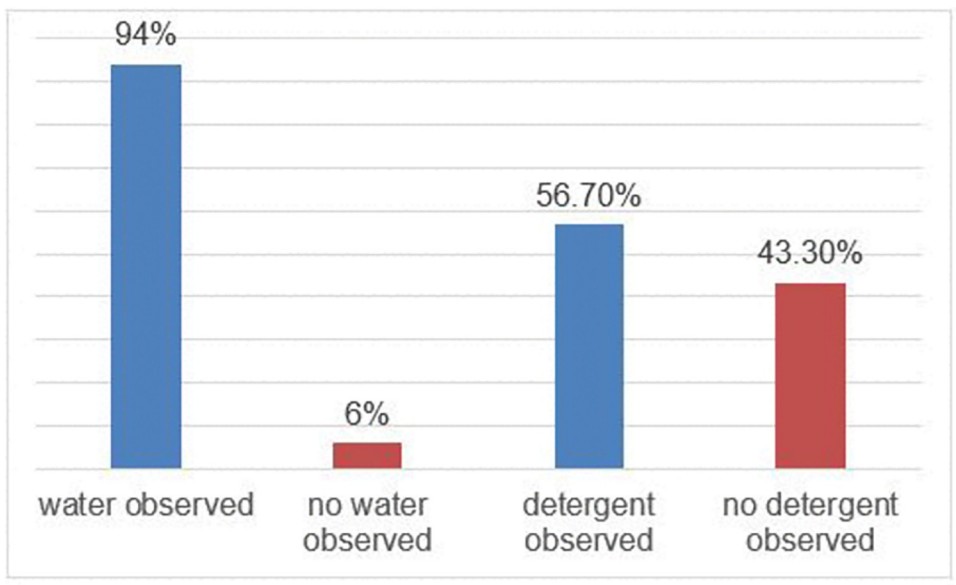

**Fig 2. Availability of water and detergents for the hand washing among the households that participated in the study of latrine utilization, 2022 (n = 365).**

**Table 3. Environmental factors of latrine utilization in East Wollega Zone, western Ethiopia, 2022 (n = 461).**

| Variable category | | CLTSH ever implemented | | Overall Frequency (%) |
|---|---|---|---|---|
| | | Yes (n = 229); frequency (%) | No (n = 232); frequency (%) | |
| Latrine has door | No | 124(54.1) | 155 (66.8) | 279 (60.5) |
| | Yes | 105 (45.9) | 77 (33.2) | 182 (39.5) |
| Functional latrine | No | 94 (41) | 108 (46.6) | 202 (43.8) |
| | Yes | 135 (59) | 124 (53.4) | 259 (56.2 |
| Presence and type of slab | Yes, by concrete slab | 20 (8.7) | 16 (6.9) | 36 (7.8) |
| | Yes, by earth/wood | 189 (82.5) | 193 (83.2) | 382 (82.9) |
| | No/open | 20 (8.7) | 23 (9.9) | 43 (9.3) |
| Number of times latrines were constructed since household establishment | One | 54 (23.6) | 64 (27.6) | 118 (25.6) |
| | Two | 134 (58.5) | 134 (57.8) | 268 (58.1) |
| | Three | 36 (15.7) | 31 (13.4) | 67 (14.5) |
| | 4 and above | 5 (2.2) | 3 (1.3) | 8 (1.7) |
| Source of information regarding latrine construction | Health professionals | 187 (81.7) | 200 (86.2) | 387 (83.9) |
| | Through sanitation campaign | 5 (2.2) | 5 (2.2) | 10 (2.2) |
| | Family members | 9 (3.9) | 8 (3.4) | 17 (3.7) |
| | Mass media | 8 (3.5) | 8 (3.4) | 16 (3.5) |
| | Neighborhoods | 11 (4.8) | 2 (0.9) | 13 (2.8) |
| | Students | 9 (3.9) | 9 (3.9) | 18 (3.9) |
| Toilet needs maintenance | No | 82 (35.8) | 72 (31.0) | 154 (33.4) |
| | Yes | 147 (64.2) | 160 (69.0) | 307 (66.6) |
| Part of toilet that needs maintenance (n = 307) | Super structure | 16 ( | 12 () | 28 (9.1) |
| | Slab | 41 | 60 | 101 (32.9) |
| | Roof | 18 | 26 | 44 (14.3) |
| | Mixed | 69 | 60 | 129 (42) |
| | Other | 3 | 2 | 5 (1.6) |
| Distance from kebele office | Near (<30 minutes) | 25 (10.9) | 29 (12.5) | 54 (11.7) |
| | Medium (30–60 minutes) | 157 (68.6) | 145 (62.5) | 302 (65.5) |
| | Too far (>60 minutes) | 47 (20.5) | 58 (25.0) | 105 (22.8) |
| Latrine Distance from dwelling in meter | >10 | 16 (7.0) | 8 (3.4) | 24 (5.2) |
| | ≤10 | 213 (93.0) | 224 (96.6) | 437 (94.8) |
| Latrine Distance from water source in meter | ≤15 | 12 (5.2) | 9 (3.9) | 21 (4.6) |
| | >15 | 217 (94.8) | 223 (96.1) | 440 (95.4) |
| Duration since toilet owned | ≤2 years | 111 (48.5) | 123 (53.0) | 234 (50.8) |
| | >2years | 118 (51.5) | 109 (47.0) | 227 (49.2) |
| Distance from HI in minutes | Near (≤60) | 126 (55.0) | 140 (60.3) | 266 (57.7) |
| | Medium (61–120) | 92 (40.2) | 80 (34.5) | 172 (37.3) |
| | Far (>120) | 11 (4.8) | 12 (5.2) | 23 (5.0) |
| Frequency of supervision by health professionals in a month | No supervision | 12 (5.2) | 7 (3.0) | 19 (4.1) |
| | One | 53 (23.1) | 45 (19.4) | 98 (21.3) |
| | Two | 149 (65.1) | 159 (68.5) | 308 (66.8) |
| | Three and above | 15 (6.6) | 21 (9.1) | 36 (7.8) |
| Frequency of supervision by local leaders in a week | No supervision | 22 (9.6) | 22 (9.5) | 44 (9.5) |
| | One | 155 (67.7) | 142 (61.2) | 297 (64.4) |
| | Two | 52 (22.7) | 68 (29.3) | 120 (26.0) |
| Hand washing facility | Yes | 179 (78.2) | 186 (80.2) | 365 (79.2) |
| | No | 50 (21.8) | 46 (19.8) | 96 (20.8) |

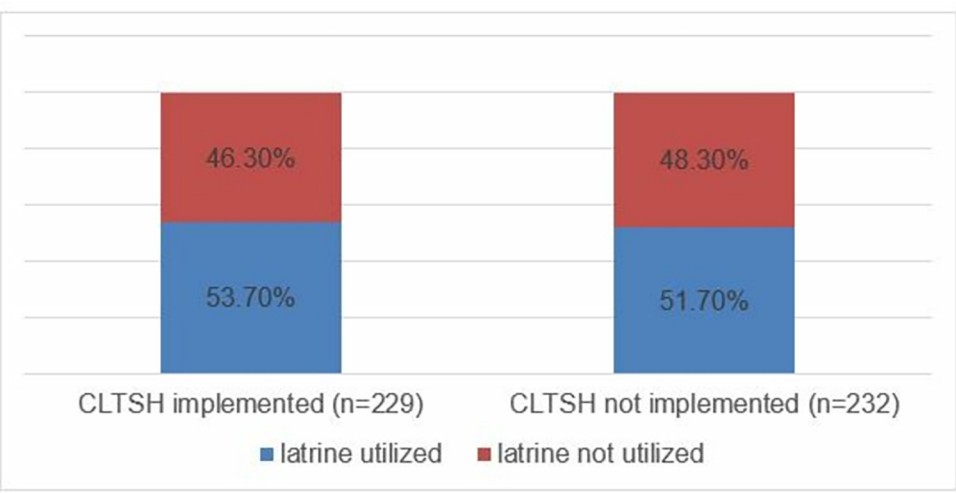

**Fig 3. Comparison of latrine utilization in CLTSH implemented and non-implemented districts of East Wollega Zone, Western Ethiopia, 2022.**

Nonetheless, only five factors such as having children <5 years, cleaning of toilet, frequency of latrine construction, maintenance need of the toilet, and distance from kebele office maintained a significant association in multivariable logistic regression (Table 4).

For the CLTSH implemented district, 17 variables such as religion, age 36–40 as compared to 18–25 years, family size, educational status of the respondent, educational status of the mother, having of educated/school children, income, having of children <5 years, occupation of the mother, cleaning toilet, motivation/reason to construct latrine, frequency of latrine construction, maintenance need, distance from health institution in minutes, distance from kebele office, latrine distance from dwelling in meter and hand washing facility associated with latrine utilization in bivariable logistic regression.

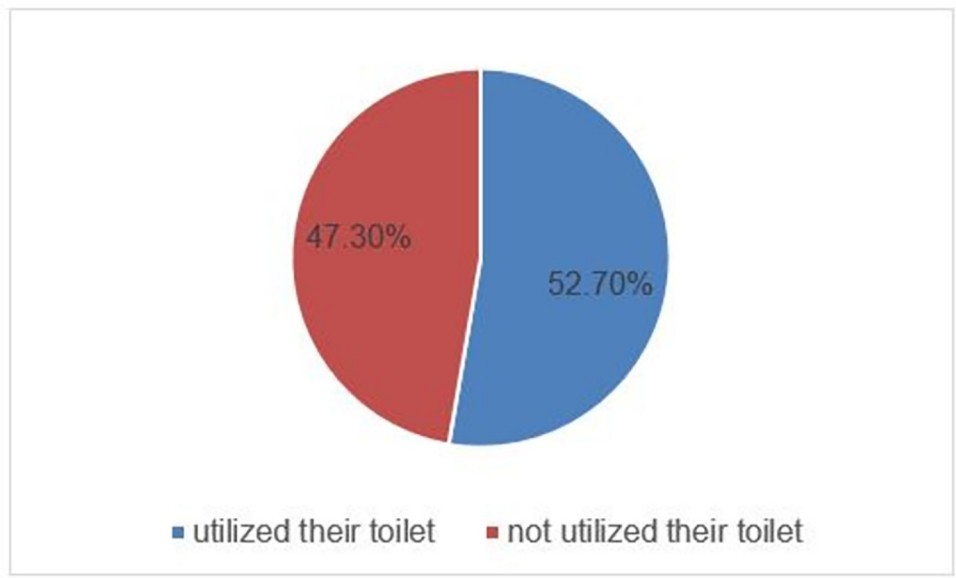

**Fig 4. Utilization of latrine among the households that participated in the study of latrine utilization and associated factors in East Wollega, Western Ethiopia, 2022 (n = 461).**

**Table 4. Factors associated with latrine utilization among CLTSH non-implemented district in East Wollega, Western Ethiopia, 2022 (n = 232).**

| Variables | Category | Latrine utilized | | | |
|---|---|---|---|---|---|
| | | No = N | Yes = N | P-value | AOR with 95%CI |
| Have children <5 years | No | 27 | 79 | 0.001* | 5.194 (1.880, 14.354) |
| | Yes | 85 | 41 | Ref | Ref |
| Clean toilet | Not clean | 88 | 27 | | |
| | Clean | 24 | 93 | 0.000* | 15.102 (5.893, 38.699) |
| Number of times latrines were constructed since household establishment | One | 29 | 35 | Ref | Ref |
| | Two | 73 | 61 | 0.333 | 1.648 (0.600, 4.528) |
| | Three | 9 | 22 | 0.019* | 5.754 (1.327, 24.937) |
| | > = 4 | 1 | 2 | 0.379 | 0.161 (0.003, 9.375) |
| Maintenance need | No | 20 | 52 | 0.006* | 4.221 (1.498, 11.890) |
| | Yes | 92 | 68 | Ref | Ref |
| Distance from kebele office | Near (<30') | 12 | 17 | 0.001* | 13.805 (2.818, 67.645) |
| | Medium(30–60") | 61 | 84 | 0.001* | 7.188 (2.190, 23.594) |
| | Too far (>60') | 39 | 19 | Ref | Ref |

AOR = adjusted odds ratio, CI = confidence interval, * p-value<0.05, Ref = Reference, N = Number

Nonetheless, only 9 factors like religion, educational status of the mother, having of children <5 years, cleaning of toilet, frequency of latrine construction, maintenance need, motivation/reason to construct latrine, latrine distance from dwelling and distance from kebele office maintained the significant association in multivariable logistic regression (Table 5).

Eighteen (18) variables showed an association with latrine utilization in bivariable logistic regression during the overall analysis of the sampled households. Age of the household head, religion, family size, educational status of the respondent, educational status of the mother, having school children/educated children, absence of children <5 years, occupation of household head, occupation of the mother, toilet cleaning, motivation/reason to construct latrine, knowledge about disease preventability of latrine usage, frequency of latrine construction, maintenance need of toilet, distance from health institution in minutes, distance from kebele office, latrine distance from dwelling in meter and hand washing facility.

However, only ten (10) variables showed significant association in the multivariable logistic regression. religion, educational status of the respondent, occupation of the mother, absence of children <5 years, toilet cleaning, frequency of latrine construction, current maintenance need of the toilet, distance from health institution in minutes, distance from kebele office, and latrine distance from the dwelling.

Those who were in other religious categories (Catholic, Muslim, and Adventist) were 85.1% (AOR = 0.149;95%CI:0.044, 0.506) less likely to utilize latrine than protestant followers. The odds of latrine utilization was 3.86 (AOR = 3.861;95%CI:1.642, 9.077) times more likely for the households whose heads can write and read as compared to those who are unable to read and write. Mothers who were housewives in occupation were two times (AOR = 2.440;95% CI:1.028, 5.792) more likely to utilize latrines than those who were farmers.

Households in which there are no under-five children utilize latrines 4.7 (AOR = 4.724;95% CI:2.313, 9.648) times more likely than their counterparts. The odds of latrine utilization was 10.6 times more likely (AOR = 10.662;95%CI:5.571, 20.403) for those households who clean their toilet when compared with their counterparts. Households that constructed toilets three times utilized latrines 6 times more likely (AOR = 6.441;95%CI:2.203,18.826) as compared to the first-time toilet construction. Similarly, the odds of household latrine utilization is 6 times

**Table 5. Factors associated with latrine utilization among CLTSH-implemented district in East Wollega, Western Ethiopia, 2022 (n = 229).**

| Variables | Category | Latrine utilized | | | |
|---|---|---|---|---|---|
| | | No = N | Yes = N | P-value | AOR with 95%CI |
| Religion | Protestant | 33 | 42 | Ref | Ref |
| | Orthodox | 58 | 75 | 0.116 | .286 (0.060, 1.364) |
| | Others (Catholic, Muslim and Adventist) | 15 | 6 | 0.008* | 0.053 (0.006, 0.463) |
| Educational status of the mother | Cannot read and write | 33 | 57 | Ref | Ref |
| | can read and write | 29 | 30 | 0.867 | 0.883 (0.205, 3.799) |
| | Primary school (1–8) | 36 | 30 | 0.004* | 0.095 (0.019, 0.479) |
| | High school and above | 8 | 6 | 0.435 | 0.148 (0.001, 17.950) |
| Have children <5 years | No | 33 | 78 | 0.013* | 5.030 (1.399, 18.081) |
| | Yes | 73 | 45 | Ref | Ref |
| Clean toilet | Not clean | 86 | 30 | Ref | Ref |
| | Clean | 20 | 93 | 0.000* | 15.152 (3.884, 59.105) |
| Motivation/reason to construct latrine | Self-initiation | 5 | 23 | 0.004* | 0.030 (0.003, 0.324) |
| | Kebele leaders and peers | 24 | 28 | 0.282 | 0.414 (0.083, 2.064) |
| | Primary health workers | 77 | 72 | Ref | Ref |
| Number of times latrines were constructed since household establishment | One | 23 | 31 | Ref | Ref |
| | Two | 74 | 60 | 0.057 | 3.789 (0.962, 14.926) |
| | Three | 8 | 28 | 0.003* | 27.698 (3.131, 244.995) |
| | > = 4 | 1 | 4 | 0.461 | 3.948 (0.103, 151.681) |
| Maintenance need | No | 17 | 65 | 0.000* | 38.391 (6.016, 245.002) |
| | Yes | 89 | 58 | Ref | Ref |
| Distance from kebele office | Near (<30') | 7 | 18 | 0.001* | 48.178 (4.761, 487.515) |
| | Medium(30–60') | 73 | 84 | 0.192 | 2.742 (0.603, 12.464) |
| | Too far (>60') | 26 | 21 | Ref | Ref |
| Latrine Distance from dwelling | >10 meter | 13 | 3 | Ref | Ref |
| | ≤10 meter | 93 | 120 | 0.018* | 30.751 (1.786, 529.593) |

AOR = adjusted odds ratio, CI = confidence interval, * p-value<0.05, Ref = Reference, N = Number

higher (AOR = 6.446;95%CI:3.023,13.744) for the latrine that doesn't need maintenance currently when compared to its counterpart. In another way, when the distance of the household from the health institutions increases by one minute, the odds of latrine utilization decrease by 1.3% (AOR = 0.987;95%CI:0.978, 0.996). Those households that were near to kebele office had 6.5 times more odds of latrine utilization (AOR = 6.478;95%CI:2.137,19.635) as compared to those too far from the kebele office. Besides, households located at a medium distance from the kebele office had 4 times more odds of latrine utilization (AOR = 4.163;95%CI:1.818, 9.531). Households whose latrine is ≤10 meters from the dwelling had 2.1 times increased odds of latrine utilization (AOR = 11.656;95%CI:2.108, 64.440) as compared to their counterparts (Table 6).

## Discussion

The prevalence and associated factors of latrine utilization among households in the East Wollega Zone were assessed in this study. Accordingly, the prevalence of latrine utilization was

**Table 6. Factors associated with overall latrine utilization in East Wollega, Oromia, Ethiopia, 2022 (n = 461).**

| Variables | Category | Latrine utilized | | | |
|---|---|---|---|---|---|
| | | No = N | Yes = N | P-value | AOR with 95%CI |
| Religion | Protestant | 73 | 78 | Ref | Ref |
| | Orthodox | 120 | 154 | 0.508 | 0.786 (0.385, 1.604) |
| | Others (Catholic, Muslim & Adventist) | 25 | 11 | 0.002* | 0.149 (0.044, 0.506) |
| Educational status of the household head | Cannot read and write | 59 | 81 | Ref | Ref |
| | can read and write | 47 | 74 | 0.002* | 3.861 (1.642, 9.077) |
| | Primary school (1–8) | 98 | 64 | 0.930 | 0.965 (0.433, 2.151) |
| | High school (9–12) | 11 | 12 | 0.142 | 3.111 (0.683, 14.165) |
| | college/university | 3 | 12 | 0.073 | 5.163 (0.858, 31.080) |
| Have children <5 years | No | 60 | 157 | 0.000* | 4.724 (2.313, 9.648) |
| | Yes | 158 | 86 | Ref | Ref |
| Occupation of the mother | Farmer | 157 | 82 | Ref | Ref |
| | Housewife | 51 | 135 | 0.043* | 2.440 (1.028, 5.792) |
| | daily laborer | 7 | 8 | 0.756 | 0.592 (0.022, 16.104) |
| | Others (Merchant, government and NGO employee) | 3 | 18 | 0.510 | 0.363 (0.018, 7.404) |
| Clean toilet | Not clean | 174 | 57 | Ref | Ref |
| | Clean | 44 | 186 | 0.000* | 10.662 (5.571, 20.403) |
| Number of times latrines were constructed | One | 52 | 66 | Ref | Ref |
| | Two | 147 | 121 | 0.201 | 1.608 (0.777, 3.328) |
| | Three | 17 | 50 | 0.001* | 6.441 (2.203, 18.826) |
| | 4 and above | 2 | 6 | 0.417 | 2.464 (0.279, 21.733) |
| Maintenance need | No | 37 | 117 | 0.000* | 6.446 (3.023, 13.744) |
| | Yes | 181 | 126 | Ref | Ref |
| Distance from HI in minutes | | | | 0.004* | 0.987 (0.978, 0.996) |
| Distance from kebele office | Near (<30') | 19 | 35 | 0.001* | 6.478 (2.137, 19.635) |
| | Medium(30–60') | 134 | 168 | 0.001* | 4.163 (1.818, 9.531) |
| | Too far (>60') | 65 | 40 | Ref | Ref |
| Latrine Distance from dwelling | >10 Meter | 20 | 4 | Ref | Ref |
| | ≤10 Meter | 198 | 239 | 0.005* | 11.656 (2.108, 64.440) |

* p-value <0.05, AOR-Adjusted odds ratio, CI-confidence interval, N-Number, ref-reference category.

found to be 52.7% (95%CI:48%, 57.3%). It was 51.7% in CLTSH non-implemented villages while 53.7% in CLTSH implemented villages. The value is almost similar to the national prevalence-50.02% [22]. It is also consistent with the study conducted in the Laelai Maichew district of Tigray in which the latrine utilization of 54.9% among CLTSH implemented and overall latrine utilization of 46.8%-58.9% was observed [35, 42]. However, the latrine utilization among CLTSH non-implemented villages in Laelai Maichew district, Tigray- 38.7% [35] was found to be lower than our finding. The current finding is also higher than the result of latrine utilization from the study in South East Zone of Tigray-37.6% [43], and the study in Chencha District, Southern Ethiopia- 31.08% [26]. The discrepancy could result from the difference in socio-economic status, and difference in study settings, and periods.

The overall latrine utilization prevalence in our study is lower than the reports of latrine utilization in Sebeta Hawas-68% [23], Mehal Meda town-90% [24], semi-urban areas of North Eastern Ethiopia-71.8% [37], Denbia district-86.8% [25], Aneded district-63% [44], Hulet Ejju Enessie Woreda-60.7% [41], Chencha district-60% [26], Wondo Genet district, South Ethiopia-83.1% [45], and Hawassa town [38]. The latrine utilization among CLTSH-implemented

villages in this study was also lower than the one in Gurage Zone's Community-Led Total Sanitation and Hygiene implemented kebeles-65.8% [46]. The observed difference might be due to the difference in the way of latrine utilization measurement, and study settings as some of the above studies were conducted in urban and semi-urban areas [23, 24, 37]. In this regard, those households that had open pits without any form of the slab were categorized as not utilizing the latrine in our study because such latrines are unlikely to break the chain of disease transmission [40].

Those households who were followers of the protestant religion were found to utilize their toilet more likely than those in other religious categories (Catholic, Muslim, and Adventist). As to the authors' knowledge, a similar finding was not observed in the literature reviewed. However, the possible explanation is that the majority of the study households were protestant followers. Followers of the Catholic, Muslim, and Adventist religions were few among those studied households. Hence the variation observed might be attributed to location not religion itself [47].

Household heads who were able to write and read used their latrine about 3.86 times more likely than their counterparts. Similar results were reported from the studies conducted in SebetaHawas Woreda [23], Laelai Maichew Woreda [42], Chencha [26], and Gurage zone [46]. This is explained by the fact that education can increase the knowledge about the diseases related to improper disposal of human wastes and lead to changes in latrine utilization behavior [22].

Occupation of the mother was another factor that was significantly associated with latrine utilization in this study. Households with housewives were about two times more likely to utilize latrines than those whose mothers were farmers in occupation. This finding agrees with the findings of the studies conducted in Denbia [25] and Laelai Maichew [42] districts. The observed association between latrine utilization and the mother's occupation might be attributed to the large proportion of housewife participants in this study. Because, the majority of the mothers in the study households were housewives. In another way, housewives might have time to keep the latrine clean and make it ready for utilization. They might easily pick and dispose of the children's feces into the toilet since they are around the home [25].

Absence of the under-five children in the household is one of the factors that increased the proper utilization of latrines compared to those having children under five years of age. This in line with the study conducted in Denbia district of North West Ethiopia in which the extent of latrine utilization was 62.1% less likely for households having ≤5 years of children [25]. The possible reason is that children under five years could not use the latrine by themselves and they might defecate in the compound [25] which could, in turn, lead to the detection of stool in the compound during observation of the latrine utilization.

In this study, households that clean their toilet and have clean toilets had a high probability of utilizing them than their counterparts. This is relevant to the studies conducted in Denbia [25], Aneded [44], Takussa [48] districts, and Gurage [46] Zone. Although it is difficult to ascertain the direction of the association, households that care for and clean their toilet might utilize it. It might be attributed to the fact that cleanness of the latrine motivates the utilization behavior whereas unhygienically handled toilets might be disgusting to use [49, 50].

The odds of latrine utilization was about 6 times more likely for those households that constructed a toilet for the third time as compared with the first-ever toilet construction. A related finding was reported from the study conducted in Denbia district [25]. The possible explanation for the observed association is that frequent latrine construction by itself might arise from the knowledge of latrine benefits and continuous use. Frequent latrine construction might be an indication of persistent latrine use and well-developed latrine behavior where as new users of latrine might have unconfirmed knowledge and behavior [25].

The odds of household latrine utilization is 6 times higher for the latrine that doesn't need maintenance currently when compared to its counterpart. This finding is in line with the studies in Guraghe zone [46] and Awabel district [51]. It might be difficult to utilize the latrine which needs maintenance. Latrine whose slabs and superstructures are well-maintained promotes its utilization by ensuring the privacy and safety of the users while the reverse is true for unmaintained latrines [46].

The distance from the health institution is inversely associated with latrine utilization in this study. When the household distance from the health institution increases, latrine utilization decreases. In another way, the odds of latrine utilization for near and medium households from the kebele office were 6.5 and 4 times more likely than those too far from the kebele office respectively. This is supported by another study conducted in Awabel District, Northwest Ethiopia. The households near woreda health center had more chance of latrine utilization [51]. This might be attributed to the accessibility of health institutions and kebele offices that in turn could expose the households to information, follow-up, and supervision [51].

Households whose latrine is found within ten (10) meters distance from the dwelling had 2.1 times increased odds of latrine utilization as compared to their counterparts. A consistent result was reported from the study in Awabel District, Northwest Ethiopia [51]. This is because a distant latrine location can reduce the chance of utilization particularly during the night time [52]. However, it is recommended for the latrine to be not too near (<6 meter) to the house [40].

Unexpectedly, there is no significant difference in latrine utilization between CLTSH-implemented and non-implemented villages. This result is inconsistent with the finding of the studies in Laelai Maichew district and Hawassa town [35, 38]. The possible reason for the discrepancy could be that currently CLTSH is not in active implementation in the current study area as it was in the initial time. The status of CLTSH implementation described in this study is ever implementation of CLTSH not necessarily the current implementation of CLTSH as this approach has faced a strong implementation challenges like unimproved sanitation performance, not implemented as intended, and absence of post-trigger follow-up [53].

The strengths of this study are that latrine utilization and the associated factors are separately determined and compared between CLTSH-implementing and non-implementing villages. However, it has its limitations. The first limitation is the cross-sectional nature of the study design which couldn't ascertain the direction of association between the independent factors and latrine utilization. Secondly, only two districts were involved which could affect the generalizability of the findings to whole districts in the East Wollega Zone.

## Conclusion

The latrine utilization in this study was found to be lower as compared to many other studies in different parts of the country and the open defecation-free mobilization. Religion, educational status of the household head, occupation of the mother, absence of children <5 years, toilet cleaning, frequency of latrine construction, maintenance need of a toilet, distance from health institution, distance from kebele office, and latrine distance from dwelling were the factors that found to affect latrine utilization. Hence, households need to maintain their toilets on time, construct them proactively before they become out of service, clean their toilets frequently and make them ready for utilization at all times. Besides, better if they construct a latrine at an accessible distance from the dwelling in a manner that promotes its utilization and that does not contaminate the groundwater sources. In addition to this, the adults in the household have to demonstrate the latrine utilization for children, and safely dispose of the feces of small children who cannot use the toilet independently. Health workers and health

officials including health extension workers should visit the households, teach them about toilet utilization and its advantage, and encourage them to appropriately utilize their toilets including the safe disposal of children's feces. The local kebele leaders have to supervise the status of toilet construction, and utilization among the households of their catchment and urge them to prepare and utilize the prepared toilet effectively. Religious leaders should also encourage the construction and utilization of latrine as it prevents the chain of disease transmission and promotes the health of the society.

## Supporting information

**S1 Checklist. STROBE statement—checklist of items that should be included in reports of observational studies.**
(DOCX)

**S1 Data. Dataset analyzed in preparing this article.**
(SAV)

## Acknowledgments

Our thanks go to Wollega University for giving us this opportunity to conduct scientific research on the problems of the community. We would like to extend the special thanks to the East Wollega zonal health department, health offices in the study districts, and primary health care unit directors for giving us valuable information and permission to conduct the study. Finally, we thank the data collectors, supervisors, and study participants for their cooperation.

## Author Contributions

**Conceptualization:** Adisu Tafari Shama, Dufera Rikitu Terefa, Edosa Tesfaye Geta, Melese Chego Cheme, Adisu Ewunetu Desisa, Dejene Seyoum Gebre.

**Data curation:** Adisu Tafari Shama, Edosa Tesfaye Geta, Melese Chego Cheme, Bayise Biru, Jira Wakoya Feyisa, Adisu Ewunetu Desisa, Dejene Seyoum Gebre.

**Formal analysis:** Adisu Tafari Shama, Edosa Tesfaye Geta, Melese Chego Cheme, Bayise Biru, Jira Wakoya Feyisa, Matiyos Lema, Adisu Ewunetu Desisa, Bikila Regassa Feyisa.

**Funding acquisition:** Adisu Tafari Shama, Edosa Tesfaye Geta, Dejene Seyoum Gebre.

**Investigation:** Adisu Tafari Shama, Dufera Rikitu Terefa, Adisu Ewunetu Desisa, Dejene Seyoum Gebre.

**Methodology:** Adisu Tafari Shama, Dufera Rikitu Terefa, Edosa Tesfaye Geta, Adisu Ewunetu Desisa.

**Project administration:** Adisu Tafari Shama, Dejene Seyoum Gebre.

**Resources:** Adisu Tafari Shama, Dufera Rikitu Terefa.

**Software:** Adisu Tafari Shama, Dufera Rikitu Terefa, Edosa Tesfaye Geta, Melese Chego Cheme, Bayise Biru, Jira Wakoya Feyisa, Matiyos Lema, Bikila Regassa Feyisa.

**Supervision:** Adisu Tafari Shama, Matiyos Lema.

**Validation:** Adisu Tafari Shama, Edosa Tesfaye Geta, Bayise Biru, Jira Wakoya Feyisa, Bikila Regassa Feyisa.

**Visualization:** Adisu Tafari Shama, Edosa Tesfaye Geta, Bayise Biru, Jira Wakoya Feyisa, Matiyos Lema, Bikila Regassa Feyisa.

**Writing – original draft:** Adisu Tafari Shama, Dufera Rikitu Terefa, Melese Chego Cheme, Bayise Biru, Jira Wakoya Feyisa, Matiyos Lema, Bikila Regassa Feyisa, Dejene Seyoum Gebre.

**Writing – review & editing:** Adisu Tafari Shama, Melese Chego Cheme, Bayise Biru, Jira Wakoya Feyisa, Matiyos Lema, Adisu Ewunetu Desisa, Bikila Regassa Feyisa, Dejene Seyoum Gebre.

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
