## [Decision Letter · Decision Letter 0]

26 May 2023

PONE-D-23-00656Latrine Utilization and Associated Factors Among Districts Implementing and Not-Implementing Community-Led Total Sanitation and Hygiene in East Wollega, Western Ethiopia: A Comparative Cross-Sectional StudyPLOS ONE

Dear Dr. Shama,

Thank you for submitting your manuscript to PLOS ONE. After careful consideration, we feel that it has merit but does not fully meet PLOS ONE’s publication criteria as it currently stands. Therefore, we invite you to submit a revised version of the manuscript that addresses the points raised during the review process.

We look forward to receiving your revised manuscript.

Kind regards,

Mohammed Hasen Badeso, MPH in Field Epidemiology

Academic Editor

PLOS ONE

4. We note that you have referenced (WSP (Water safety plan). Desk review on Economics of Sanitation (ESI) for Ethiopia. 2015; unpublished.]) which has currently not yet been accepted for publication. Please remove this from your References and amend this to state in the body of your manuscript: as detailed online in our guide for authors

http://journals.plos.org/plosone/s/submission-guidelines#loc-reference-style "

Additional Editor Comments:

Dear Author(s),

Thank for submission your manuscript to PLOS ONE journal. The peer review process for your manuscript is now completed.

We would be grateful if you could address the reviewers comments in a revised manuscript and provide point-by-point response to the concerns raised. In addition, ensure that your revised manuscript conforms to the journal style available in the journal instructions for authors. It is important that your files are correctly formatted.

Reviewers' comments:

Reviewer's Responses to Questions

**Comments to the Author**

1. Is the manuscript technically sound, and do the data support the conclusions?

Reviewer #1: Yes

Reviewer #2: Yes

2. Has the statistical analysis been performed appropriately and rigorously? 

Reviewer #1: Yes

Reviewer #2: Yes

3. Have the authors made all data underlying the findings in their manuscript fully available?

Reviewer #1: Yes

Reviewer #2: Yes

4. Is the manuscript presented in an intelligible fashion and written in standard English?

Reviewer #1: Yes

Reviewer #2: Yes

5. Review Comments to the Author

Reviewer #1: The manuscript had tried to insight a good topic and I would like to appreciate the authors. However, I have some comments and questions.

Abstract

Comment 1: you didn’t provide a good justification to conduct the study under the background.

Comment 2: the result is not written in concise and clear manner as well very congested and doesn’t attract the reader.

Background

Comment 1: in the last paragraph of the background you had tried to justify the need for the study and tried to show some strengths of the study. In the paragraph you have mentioned that other studies didn’t investigate safely managed services and this study fills the gap. However, in this study nothing has been showed in the methods (especially operational definition) and result (Latrine coverage by sanitation ladder) about safely managed services. So, you better exclude this justification otherwise include the above comments in your methods and result part.

Methods and materials

Comment 1: for the qualitative study you planned and executed thematic analysis. However your result didn’t show that and the qualitative result is not presented in a good manner. As well the discussion doesn’t incorporate findings from the qualitative study. By the way, it is better to exclude the qualitative study whole, if not so try to read and read about triangulation and thematic analysis and come up with an organized result and discussion.

Comment 2: there is no need to have these paragraphs in the operational definition “The use of the latrine was assessed based on self-reporting, and the observation of proxy indicators. Accordingly” and “Finally, the household was categorized as utilized or not utilized based on the above definition”

Comment 3: as per the operational definition Latrine utilization is measured among households having an improved toilet. So what are improved latrines/toilets? ; operationalize the facilities and try to show the reader how many of the households own improved toilets in your manuscript clearly. Furthermore what is a functional toilet? This also needs to be operationalized.

Comment 4:

Result

Comment 1: almost all the tables are distorted and poorly organized; this needs to be clearly addressed.

Comment 2: what is the purpose of having this much category of age groups? I think you better categorize age not more than 4 or 5 age groups.

Comment 3: line 278 to 280, what is Squat hole cover? ; What you are talking about is a slab not a squat hole cover.

Comment4: line 290 water, sanitation and hygiene focal person better be “WaSH focal person”

Comment 5: for the qualitative study, all the second person paragraphs or ideas better be italic and should be placed separately.

Comment 6: line 297 and 302, A 34 what? How

Comment 7: the variable distance from the kebele. How near is near and how far is far? This should have to be operationalize.

Comment 8: what does the variable frequency of latrine construction means?

Comment 9: the quantitative data focuses on latrine utilization and the qualitative result showed about issues other than that. Therefore as I tried to mention earlier it would be better to exclude from the manuscript.

Discussion

Comment 1: generally almost all your justifications are not supported by evidences from other literatures. You just forward your personal opinion all over the discussion. So please try to read and incorporate other literatures.

Reviewer #2: The manuscript was well written and only have minor comments. The problem is stated clearly, and the result written in well manner. There was not too much grammatical error in the manuscript and all data incorporated in the result part.

6. PLOS authors have the option to publish the peer review history of their article (what does this mean?). If published, this will include your full peer review and any attached files.

Reviewer #1: No

Reviewer #2: **Yes: **Dinku Mekbib

---

## [Author Response · Author response to Decision Letter 0]

11 Jun 2023

Authors’ point-by-point response to the editor and Reviewers comments 

First, the authors would like to thank the editor and reviewers for reviewing our manuscript and providing us the constructive comments that help us to enrich the document. We tried to address all the comments and made necessary modifications and clarifications based on the comments and suggestions provided.

Comments by editor

We opened those links and checked the alignment of our manuscript with the formats and styles required.

The data set is uploaded

Accepted.

4. We note that you have referenced (WSP (Water safety plan). Desk review on Economics of Sanitation (ESI) for Ethiopia. 2015; unpublished.]) which has currently not yet been accepted for publication. Please remove this from your References and amend this to state in the body of your manuscript: as detailed online in our guide for authors

http://journals.plos.org/plosone/s/submission-guidelines#loc-reference-style "

Corrected

5. Review Comments to the Author

Abstract

Comment 1: you didn’t provide a good justification to conduct the study under the background. Amended in this revised manuscript.

Comment 2: the result is not written in concise and clear manner as well very congested and doesn’t attract the reader. Amended.

Background

Comment 1: in the last paragraph of the background you had tried to justify the need for the study and tried to show some strengths of the study. In the paragraph you have mentioned that other studies didn’t investigate safely managed services and this study fills the gap. However, in this study nothing has been showed in the methods (especially operational definition) and result (Latrine coverage by sanitation ladder) about safely managed services. So, you better exclude this justification otherwise include the above comments in your methods and result part. 

We accepted this suggestion and removed the description about ‘safely managed services’ from the manuscript.

Methods and materials

Comment 1: for the qualitative study you planned and executed thematic analysis. However your result didn’t show that and the qualitative result is not presented in a good manner. As well the discussion doesn’t incorporate findings from the qualitative study. By the way, it is better to exclude the qualitative study whole, if not so try to read and read about triangulation and thematic analysis and come up with an organized result and discussion. 

We accepted this and removed the qualitative part.

Comment 2: there is no need to have these paragraphs in the operational definition “The use of the latrine was assessed based on self-reporting, and the observation of proxy indicators. Accordingly” and “Finally, the household was categorized as utilized or not utilized based on the above definition” 

Amended.

Comment 3: as per the operational definition Latrine utilization is measured among households having an improved toilet. So what are improved latrines/toilets? ; operationalize the facilities and try to show the reader how many of the households own improved toilets in your manuscript clearly. Furthermore what is a functional toilet? This also needs to be operationalized. 

Accepted and operationalized in this revised manuscript..

Comment 4:

Result

Comment 1: almost all the tables are distorted and poorly organized; this needs to be clearly addressed. 

The tables are modified

Comment 2: what is the purpose of having this much category of age groups? I think you better categorize age not more than 4 or 5 age groups. 

We recategorized the age based on the reviewer’s suggestion.

Comment 3: line 278 to 280, what is Squat hole cover? ; What you are talking about is a slab not a squat hole cover.

Thank you. We corrected the variable as “presence of slabs for the latrine”

Comment4: line 290 water, sanitation and hygiene focal person better be “WaSH focal person” 

Comment 5: for the qualitative study, all the second person paragraphs or ideas better be italic and should be placed separately.

These comments were addressed as we already removed the qualitative part from this manuscript.

Comment 6: line 297 and 302, A 34 what? How

Comment 7: the variable distance from the kebele. How near is near and how far is far? This should have to be operationalize. 

Thank you! We specified this as near if walking distance of <30 minutes, medium (30-60 minutes), and too far (>60 minutes).

Comment 8: what does the variable frequency of latrine construction means? We tried to make this variable clear by modifying as “number of times latrines constructed since household establishment”

Comment 9: the quantitative data focuses on latrine utilization and the qualitative result showed about issues other than that. Therefore as I tried to mention earlier it would be better to exclude from the manuscript. We accepted this comment.

Discussion

Comment 1: generally almost all your justifications are not supported by evidences from other literatures. You just forward your personal opinion all over the discussion. So please try to read and incorporate other literatures.

We accepted this comment, modified the discussion and supported our justification by references. 

Reviewer #2: The manuscript was well written and only have minor comments. The problem is stated clearly, and the result written in well manner. There was not too much grammatical error in the manuscript and all data incorporated in the result part.

Thank you! I hope those minor comments have been addressed in this revised version.

---

## [Editor Report · Decision Letter 1]

28 Jun 2023

Latrine Utilization and Associated Factors Among Districts Implementing and Not-Implementing Community-Led Total Sanitation and Hygiene in East Wollega, Western Ethiopia: A Comparative Cross-Sectional Study

PONE-D-23-00656R1

Dear Author(s),

We’re pleased to inform you that your manuscript has been judged scientifically suitable for publication and will be formally accepted for publication once it meets all outstanding technical requirements.

Kind regards,

Mohammed Hasen Badeso, MPH in Field Epidemiology

Academic Editor

PLOS ONE
---

## [Editor Report · Acceptance letter]

3 Jul 2023

PONE-D-23-00656R1 

Latrine utilization and associated factors among districts implementing and not-implementing community-led total sanitation and hygiene in East Wollega, Western Ethiopia: A comparative cross-sectional study 

Dear Dr. Shama:

I'm pleased to inform you that your manuscript has been deemed suitable for publication in PLOS ONE. Congratulations! Your manuscript is now with our production department. 

Kind regards, 

on behalf of

Mr Mohammed Hasen Badeso 

Academic Editor

PLOS ONE